# Resilience Analysis of Australian Electricity and Gas Transmission Networks

**Shriram Ashok Kumar, Maliha Tasnim, Zohvin Singh Basnyat, Faezeh Karimi and Kaveh Khalilpour *** 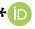

Faculty of Engineering and IT, University of Technology Sydney, Sydney, NSW 2007, Australia;
shriram.ashokkumar@alumni.uts.edu.au (S.A.K.); maliha.tasnim@outlook.com (M.T.);
basnyatzs@gmail.com (Z.S.B.); faezeh.karimi@uts.edu.au (F.K.)
**\*** Correspondence: kaveh.khalilpour@uts.edu.au

**Abstract:** Given they are two critical infrastructure areas, the security of electricity and gas networks is highly important due to potential multifaceted social and economic impacts. Unexpected errors or sabotage can lead to blackouts, causing a significant loss for the public, businesses, and governments. Climate change and an increasing number of consequent natural disasters (e.g., bushfires and floods) are other emerging network resilience challenges. In this paper, we used network science to examine the topological resilience of national energy networks with two case studies of Australian gas and electricity networks. To measure the fragility and resilience of these energy networks, we assessed various topological features and theories of percolation. We found that both networks follow the degree distribution of power-law and the characteristics of a scale-free network. Then, using these models, we conducted node and edge removal experiments. The analysis identified the most critical nodes that can trigger cascading failure within the network upon a fault. The analysis results can be used by the network operators to improve network resilience through various mitigation strategies implemented on the identified critical nodes.

**Keywords:** network science; energy networks; cascading failure; percolation theory; scale-free network; Barabási-Albert model

## 1. Introduction

### 1.1. The Resilience of Energy Networks

Power grids are considered one of the most complex networks in the modern era. These networks connect power generators to end-users through transmission and distribution lines over long distances [1]. The complexity of power networks is also increasing due to the transformation that these networks are undergoing due to factors such as renewable energy uptake and decentralisation. Given their role as critical societal infrastructure, maintaining a high level of resilience in these networks against any fault that can lead to blackouts is vital. Even if rare, such events entail catastrophic socio-economic and life threats. Hence, energy network resilience has been given greater importance in researching blackouts and obtaining more resilient power networks for preventing future devastating events.

There is evidence of large blackouts causing disruptions in different parts of the world, including developed countries such as the USA, Canada, and Germany. These events eventually affect interdependent essential infrastructure systems including energy, transportation, economy, or communication systems. For example, 50 million North Americans observed a widespread blackout on 14 August 2003, which took two days to restore power fully in some locations [2]. On 4 November 2006, a major blackout in Germany triggered a ripple effect on the power network, causing 15 million Europeans to lose access to power [3]. In northern and eastern India, major power blackouts, on 30 July and 31 July 2012, are considered two of the most extensive power outages in history, impacting 620 million people, which is half of India's population [4]. In 2016, South Australia experienced a widespread power outage triggered by a storm event that affected

850,000 people throughout the state [5]. These large blackouts are technically described as cascading failures of the power network.

Sometimes, cascading failure can be initiated by a limited number of disturbances or errors leading to a ripple effect throughout the whole network. North American Electric Reliability Corporation (NERC) defines cascading failure as "the uncontrolled successive loss of system elements triggered by an incident at any location" [6]. Studies have also shown how an extensive complex power network can disintegrate due to the collapse of a single transformer, which eventually supports cascading failure theory [7]. More recently, in May 2021, Taiwan witnessed a blackout with cascading failure mechanism triggered by a power plant in southern Taiwan which off-guarded 24 million residents affecting four million households [8]. In May 2021, an explosion at Callide Power Station in Queensland, Australia, led to a catastrophic failure affecting 470,000 customers without power [9,10]. The investigation report indicated that the fire accident at the Callide Power Station tripped other power plants, causing a domino effect across the Queensland electricity network [9,10]. Due to the occurrences of cascading events such as those described, it is crucial to study the impact of possible disturbances (random or intentional) on critical networks for developing efficient safeguards against blackouts. The growing climate change challenges, including the severe weather conditions associated with worldwide heatwaves, bushfires, and floods, are another emerging motivator of the heightened interest in the analysis of network resilience against disturbance [11,12].

### 1.2. Australian National Energy Network

This paper focuses on the Australian electricity and gas networks (refer to Figure 1) and aims to analyse their robustness against cascading failure. Here, we provide a topological description of each network.

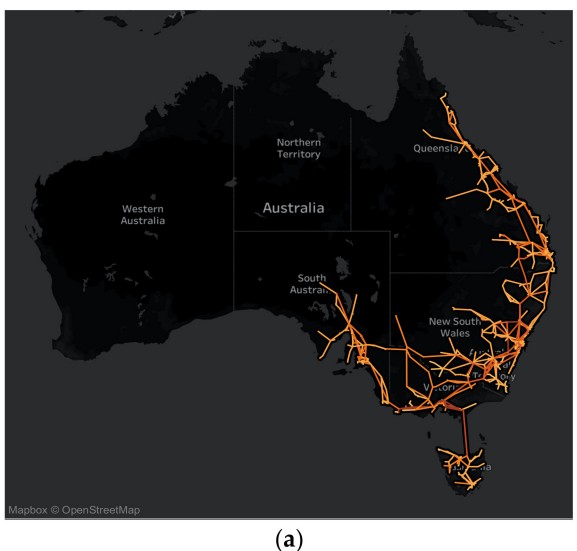　　　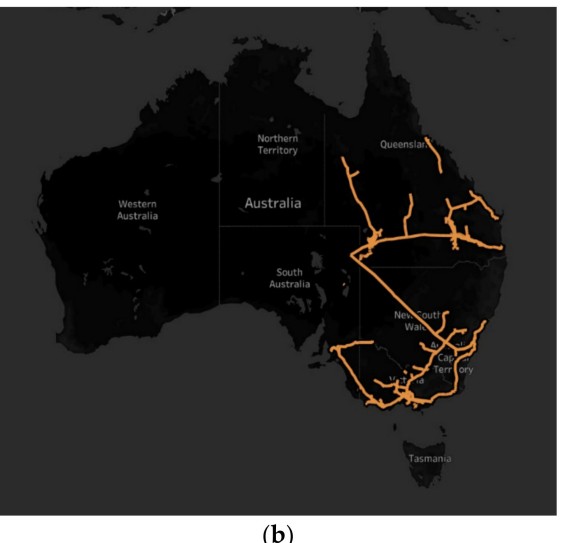

(**a**)　　　　　　　　　　　　　　　　　　　　　(**b**)

**Figure 1.** Spatial Australian (east coast) national transmission networks of (**a**) electricity lines and (**b**) natural gas pipelines.

Australian electricity network: The Australian national electricity market, operated by the Australian energy market operator (AEMO), is the longest electricity network in the world, spanning over 5000 km from northern Queensland to the states of South Australia and Tasmania as illustrated in Figure 1a [13]. The network is undergoing a notable transformation to install renewable technologies such as solar PV, wind, and pumped hydro [13]. Nevertheless, it is still a highly coal-dependent network with some old brown or black coal-fired power plants. According to Australia Institute researchers, Australia's coal-fired power plants are ageing and the country could observe frequent outages due to ageing infrastructure, extended summer weather, heatwaves, and bushfires [14]. The network has

witnessed continuous disturbances over the past decade. For instance, South Australia experienced a significant blackout in 2016. It has been reported that one distributor had to compensate 75,000 customers, a payment of more than AUD 20 million due to the blackout [15]. In 2019, due to an extreme heatwave, around 200,000 people in Victoria suffered a power supply outage [16]. Ausgrid, a distribution network provider in the state of NSW, reported a total of 2485 h of outage in the first quarter of 2019, which affected approximately 545,558 customers along with associated networks [17]. The 2021 fire at the Callide power station (coal-fired, commissioned in 1965) in Queensland is the latest major accident, affecting almost half a million customers [9,10].

Australian gas network: The gas network, also operated by the AEMO, is Australia's integrated energy system and plays a critical role in the value chain of natural gas, delivering 40% of Australia's domestic gas consumption [18]. Although significant gas network incidents are rare in Australia, the consequence of an incident can significantly affect the community. For instance, the NSW gas network performance report (2016–2017) stated that lost hours and unplanned losses were approximately 12.34 h per 1000 customers due to network disturbance [19]. In 2013, the network was highly disrupted due to a bushfire [19], and there are growing concerns about global warming impacts.

The whole gas network can collapse if there is damage to the pipeline due to inadequate maintenance, accidental damage, ageing infrastructure, weather events, sabotage, or land movement [20]. Pipelines' leakage can also cause severe health and environmental issues affecting the surrounding community, making the robust operation of this network a critical national security issue.

In summary, both networks have been facing supply-demand challenges rooted in various reasons, mainly stemming from the continuous population growth and improved quality of life through technology [21]. The addition of climate change impacts and increasing international cyber-attacks on infrastructure networks has created increasing concern over the future security of Australian gas and electricity networks.

Furthermore, immense research is needed to analyse the resilience of these networks concerning the national net-zero-emission targets toward 2050 [22]. The recent COVID-19 pandemic situation has also had short-term and medium-term impacts on these networks, increasing their operation complexity [23]. All of these necessitate a systematic resilience analysis of the two networks against any uncertain disturbance that can trigger widespread service disruption.

### 1.3. Resilience Analysis with Complex Network Theories

Various studies have proposed to increase the reliability of the energy networks against cascading failure by focusing on robustness and vulnerability measures of the networks [24]. Some studies suggest malfunction in the form of cascading failure or blackouts within power networks are caused by inadequate comprehension of the interdependencies present in the network rather than operational issues such as low investment or maintenance [25]. To grasp a better understanding of such interdependencies, the robustness epigraph of complex networks have become very popular among the scientific community [26].

Complex network theories analyse real-world networks of any type (e.g., the World Wide Web, social networks, infrastructure networks, and biological networks) by combining the concept of graph theories [27]. Similarly, the energy grids are networks comprised of many nodes (e.g., generation and load) and edges (e.g., electricity transmission lines or gas transport pipelines) as connections [28]. These networks can be characterised as complex networks by analysing their statistical characteristics, including degree, degree distribution, path length, and clustering coefficient, using graph theories [25]. Knowing these features can provide some insight for network operators to assess network robustness. For instance, Meng et al. (2004) found that the cascading failure impact of the small-world network has a unique shorter mean distance along with a higher clustering coefficient which facilitates network failure [29]. Studies such as these have motivated research efforts to identify topology features of networks before any further detailed investigation.

For example, western United States, northern Europe, and northern China electricity networks have been identified as small-world networks by comparing their clustering coefficient and path length [29,30]. Ding and Han (2006) established that China's Sichuan-Chongqing and Anhui power grids are close to the characteristics of the random network [31]. An exponential distribution has been observed in many studies [32,33], while power-law distribution has been reported for the North American Power grid networks [34]. In an informative study, Cotilla-Sanchez et al. (2012) compared three different power grids in North American regions by evaluating the statistical characteristics of a same size small-world graph, scale-free graph, and random graph [35]. The research concluded that these network distributions are more proximate to exponential distribution than power-law distribution and do not follow the characteristic of small-world or scale-free networks.

Following the former studies, this paper aims to investigate the robustness of Australian electricity and gas networks using network science theories. Our key research aims include the: (1) identification of a network model which fits best with Australian gas and electricity networks; (2) identification of robustness models which can efficiently detect the critical nodes and edges of both networks; (3) measurement of the resilience of the two networks considering random failure or targeted attacks, including nodes and edge removal experiment; and (4) comparison of the difference in topology of the two networks and their behaviour towards failure.

The following section provides a literature review of the network science theories. It also signifies the implication of the different concepts of percolation theory for checking system robustness using random graph models and scale-free networks. This will be followed by depicting the outcome of the experiments of random attacks and tolerance to cascading failures through attack measures and node and edge removal study. Last, a summary of the findings and implications has been outlined.

## 2. Literature Review

The first step in any network analysis is to identify the model that can most accurately estimate the characteristics of the given network. There are many classifications of complex network structure, such as the Erdős-Renyi (ER) stochastic network (Erdös 1959), Watts-Strogatz (WS) small-world network [36], and Barabasi-Albert (BA) scale-free network [37], which are described in this section.

### 2.1. Random Graph Models

Erdős-Rényi (ER) Model: Paul Erdős and Alfréd Rényi founded random graph theory in 1959 and over the following years, Erdős established the logicality of using probabilistic methods to tackle complex network problems [38]. Erdős's theorem claims that the existence of a graph must meet specific properties: it is a perfectly normal proclamation showing no indication of the randomness used in its substantiation [39]. According to the Erdős-Rényi (ER) model, a network can be generated by placing a number of nodes (n) and adding up edges among them in conjunction with independent probability ($p$) for each of the node pairs [40].

Gilbert-Elliott (GE) Model: As per the Erdős-Rényi model, Gilbert and Elliot used probabilistic methods, but different graph definition approaches [41]. The GE model is widely used to represent the state of a channel (G-Good, B-Bad) by analysing the errors on the channel. The model has two states: the Good state corresponds to a successful connection and the Bad state refers to a loss connection [42]. Gilbert claimed that for $N$ number of nodes, the possibility of each edge joining those pair of nodes has probability $p$. A graph with $q$ edges has the probability $p^q (1-p)^{N-q}$ where $N$ is the number of possible edges equalling $n(n-1)/2$.

### 2.2. Scale-Free Network Models

Barabási-Albert (BA) model: The key issue of random graphs is their failure to adequately predict hubs in the network, which are nodes with a very high degree, but with

low frequency (also known as the tail of a network distribution). Scale-free networks are generally dominated by a few highly connected hubs [43]. Albert et al. [44] suggested that highly interconnected nodes somehow control the behaviour of scale-free systems and their resilience. Such nodes also tend to have a higher probability of acquiring new edges. According to this model, the network considers the following two steps: (1) Growth occurs when a new node adds every time with $q$ ($\leq q_0$) links connecting $q$ existing nodes in the network. (2) Preferential attachment occurs when the connection of a new node to an existing node $i$ depends on its degree. A growth-based preferential attachment model is a highly used mechanism to explain the frequency of power-law degree distribution.

The BA model aimed to understand how a network originated in the first place and to capture how a real network grows from its origin. The limitation of the BA model is that it can provide the still status of the network in image form but fails to match the growth of the real network.

Extension of Barabási-Albert (BA) model: BA and Watts-Strogatz (WS) models have been established as pioneer models in network science considering real-life network complexity. Holme and Kim (2002) extended the BA model by including a triad formation step [26]. Klemm and Eguıluz [45] used the finite memory of nodes to construct a growing network model. Saramaki and Kaski [46] worked on an undirected scale-free model produced by random walkers. Li and Chen (LC) [47] developed a model-evolving local world network model which is applicable to the Internet and society, but the LC model fails to have a high clustering coefficient. Fan and Chen [47] developed a Multi Local World (MLW) model to describe the Internet structure with better statistical performance. The Bianconi-Barabási model is another extension of the BA model, with it being assumed that nodes with higher fitness have a greater probability of acquiring new links [48].

*2.3. Small World Network Models*

Watts-Strogatz (WS) Model: In 1998, Duncan J. Watts and Steven Strogatz introduced the small-world network model, which is based on the random graph but has a short characteristic path length with a high network cluster [49]. The model assumes that each vertex of the nodes is connected to the nearest fixed number of vertices (periodic boundary condition). A shortcut bond is created between randomly selected vertices [50]. The WS small-world network is a special model that lies just between the homogenous degree distribution and heterogeneous degree distribution models and the BA model [51]. The following are some characteristics of the WS model:

1. The model has existing cliques without hubs, a higher level of clustering coefficient;
2. The degrees of nodes follow a fat-tailed distribution;
3. The model has a smaller average path length than Erdos-Renyi networks:

$$\text{Average Path Length (APL)} = \ln n / \ln d$$

Small-world network properties: Two statistics used for detecting small-world properties are: (1) the clustering coefficient and (2) the average shortest path length. The average shortest path length (L) is defined as

$$L = \frac{1}{N(N-1)} \sum_{i=1}^{N} \sum_{j=i+1}^{N} L_{min}(i,j) \qquad (1)$$

Clustering has been found to be a common feature for many complex networks [52]. Within any network, the extent of clustering can be quantified by the clustering coefficient (C). For any node $N_i$ with $k_i$ neighbours, if $E_i$ indicates the total links between $k_i$ neighbours, then the clustering coefficient is defined as:

$$C = \frac{1}{N} \sum_{i=1}^{N} \frac{2E_i}{k_i(k_i - 1)} \qquad (2)$$

To determine a network to have small-world properties in comparison to the Erdős-Rényi random graph, the conditions require a low average path length ($L \approx L_{random\ graph}$) and high clustering coefficient ($C \gg C_{random\ graph}$) [53].

The probability distribution of the three network models: The random networks follow a Poisson distribution. The degree distribution of small-world networks (Watts-Strogatz model) follows the exponential function. It tends to decrease at a relatively higher degree with the reduction in the number of nodes. The scale-free network follows a power-law function which is quite different to the exponential function. The rate of decrease in the distribution is slower in power-law relative to the exponential function. Thereby, the degree distribution of nodes tends to decrease at slower rates in scale-free networks than the small-world networks.

## 3. Robustness Analysis of Energy Networks

Micro and macro network characteristics: In the past few decades, different fault models have been introduced to understand the mechanism of cascading failures in energy networks. The models mainly differ on their microscopic component characteristics and macroscopic topological features [27]. For instance, for the electricity network, there are three classes: static model, component cascading failure, and power network dynamics [54]. The typical examples of static models are the betweenness centrality model [26], the effective efficiency model [55], and the Motter-Lai model [56], which are used to describe the macroscopic topologies for analysing the security of energy networks. The component cascading failure model considers the component failures of a network as a reason for cascading failure rather than the specific macroscopic topologies of a network [57]. The CASCADE model and branching process model [57] are examples of component-based class. The power network dynamics category includes models such as the ORNL-PSerc-Alaska (OPA) model, hidden fault model, or Manchester model [27]. To indicate robustness, these models are described by featuring average clustering, path length, transmission efficiency, or maximum component size. Another widely studied theory for robustness analysis is percolation theory which will be discussed further in this section.

Percolation theory: Percolation is the analysis of how a network of discrete elements relates to each other. Classic percolation theory is a division of probability theory that deals with properties using random channels [58]. Here, properties include the analysis of clusters and statistics of their elements. This theory was first introduced by Broadbent and Hammersley [59], while Sahimi later evaluated the practical application of this theory in engineering problems [60]. Over the years, researchers established two models based on percolation theory, being the bond percolation model and site percolation model that will be discussed later in this section.

Percolation theory for random graph: The key drawback of random graphs in dealing with real-world networks is the unrealistic assumption that each node can attract edges with the same probability following a Poisson distribution [61]. In the real world, networks such as the World Wide Web, social networks, citation networks, and language networks show a trend of exponential degree or power-law distribution which is a non-Poisson distribution. The Watts-Strogatz model has made the percolation study more logical as it allows producing random graphs using a non-Poisson degree distribution [62]. This means with the known constant mean rate (*pc*), the probability of each node connecting with edges has no fixed pattern in terms of degree distribution within a network.

Percolation theory for scale-free networks: Schwartz et al. (2002) showed that there are direct links between many complex networks in nature and these links emulate a property that affects such networks' large-scale topology and navigability [63]. Naturally occurring networks such as the Internet, scientific collaboration, and social networks exhibit power-law or scale-free degree distributions. For instance, degree distribution $p(k)$ is referred to the probability of random nodes linked to exactly $k$ other nodes. Hence, $p(k) = Ck^{-\lambda}$ where $k \geq m$. Here, $m$ refers to minimal connectivity and $C$ refers to the normalisation factor [64]. If a large network is weakened by the arbitrary removal of a fraction ($p$) of its

nodes, then if $p$ is small, it is expected that the network will not be affected at all unless the large component of connected nodes constituting the finite fraction of the entire size of the original network is disrupted. To understand the problem of percolation, Cohen, Ben-Avraham, and Havlin (2002) claimed that when exceeding a particular threshold of dilution, the large component breaks down and the network effectively collapses [64]. For example, a scale-free network such as the Internet can still operate as a connected network after removing 95% of the nodes but the strategic removal of a small fraction (e.g., 2.5%) of the important nodes can collapse the whole network [65]. Cohen and Callaway [66,67] used percolation theory to demonstrate that if interconnected nodes are purposively removed, the whole network collapses to remote sub-networks, implying there is a strong robustness to random failure but fragility against deliberate attack.

Bond and site percolation: Addressing percolation in a network study, bond percolation refers to the probability $p$ of the existence of an edge transferring from one node to another, while site percolation describes the situation focusing on nodes rather than the edge. As such, for percolation analysis of network structure, we can assume a randomly connected network with probability $p$ for the existence of nodes (site percolation) or edge between two given nodes (bond percolation) [68]. For site percolation, probability $p = 0$ implies the existence of no cluster in the system. For bond percolation, probability $p = 0$ implies all clusters have a size of one, hence no nodes are connected in the system. Probability $p = 1$ means a single cluster representing the whole network in both models.

The function of probability $p$ is used to observe the structural change of percolation transition between these extreme configurations, which is monitored through percolation strength and the size of the largest cluster considering the percolation transition order parameter [68]. The order parameter describing the power-law growth as an exponent function is the same in both processes for analysing the distance from the critical point [69]. Critical exponents refer to the average size of finite clusters and the distribution of the cluster size, which is the singular behaviour of the observables and plays an important role to group networks in different classes through characterisation of the percolation transition properties [70]. Theoretically, there is a perfect equivalence between these two models [71] and no difference has been reported between the critical exponents of both, although their percolation thresholds could be different [67,72]. Nevertheless, more recent studies have reported inconsistencies in such site-bond universality in terms of macroscopic observable's behaviour considering anomalies in critical exponent values [68].

The percolation threshold is a fundamental concept of percolation theory which can be referred to as $pc$ [58]. It is the value of probability $p$ in which topological transition occurs and network structure shapes to a connected one from a disconnected one, and hence is an effective feature in the analyse of network robustness. The critical value of $pc$ identifies the network path for connected nodes. For example, the $pc$ for bond percolation and square 2D lattice is 0.5 [69].

K-Clique percolation theory: In a random graph, k-clique can be a subgraph where the distance between any of the two vertices is not greater than $k$ [73]. This is a popular and effective approach to identify overlapping clusters in a large scale-free real network. In k-clique percolation random graphs, $k$ is the size of total subgraphs of $k$ nodes where large-scale networks are numerically and analytically explored. Two k-cliques can be adjacent to each other when they share $k1-1$ nodes and a $k$-clique percolation cluster is equivalent to a community [74]. When there are overlapping cliques in one network, the robustness becomes very strong and thus the clique method is very efficient and interesting for studying robustness. Cliques are identified by evaluating the clustering coefficient. A higher value of the clustering coefficient indicates there is the possibility of the presence of high cliques within the network. According to the earlier theoretical discussion, a small-world network is more robust than Erdős-Rényi or BA networks as it has a high clustering coefficient.

In this report, we aim to identify how the Australian gas and electricity network is formed and what size clusters exist within the network. A network needs to have a large size of communities, as it would be strongly robust to errors. Otherwise, a deliberate attack in the critical nodes will bring catastrophes, as the network would be broken into small isolated fractions of the cluster.

## 4. The Case Study Dataset

The electricity and gas networks are the main research elements of this project. For research and experimental purposes, the extracted data was collected in XML and then converted using reptile technology. The Australian electricity network dataset, which was last revised in March 2017, was collected from the Geoscience Australia Web Service. It includes 1180 nodes (generators and substations) and 1465 edges (transmission lines).

For the purpose of this study, the main fields of the datasets are the starting and ending position of each transmission line through the network, the state where the lines belong, the voltage (kV) capacity of the line, and the longitude/latitude coordinates of the location. To contemplate a nonredundant outcome, only outsets and endings of the transmission lines are considered. While there are no transmission lines that can transfer voltage in both directions, it is evident that in some lines power passes in one direction at one point in time and transfers reversely at another point in time. As such, we are considering the undirected network here. The simplified undirected network of our study has 1180 nodes and 1465 edges, with its topology illustrated in Figure 2. The network map of Figure 2 can be defined in terms of graph $G = (n,e)$, where $n$ indicates sets of nodes in the graph including substations and generators in the electricity network and $e$ denotes the edges between the nodes.

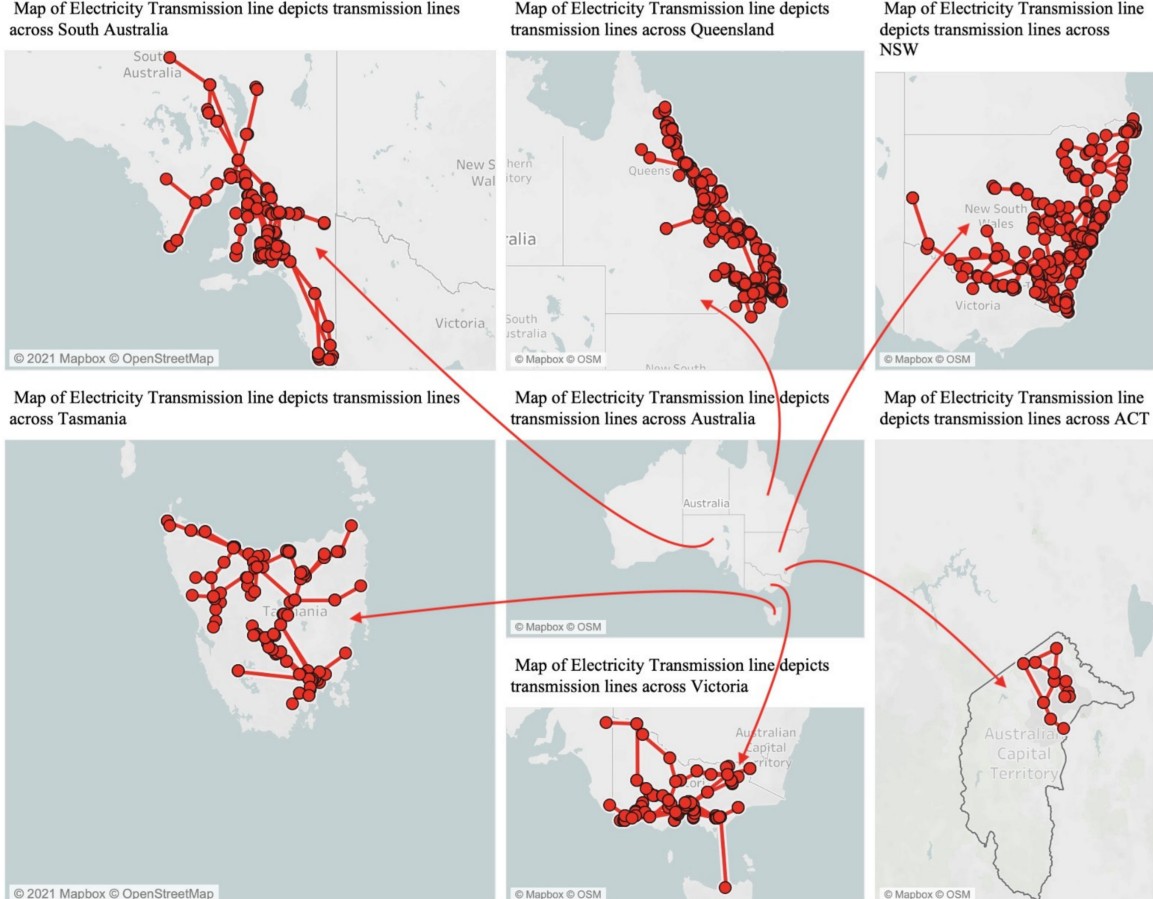

**Figure 2.** The spatial map of Australian electricity transmission lines across six states/territories.

The dataset of the Australian gas network was obtained from the Geoscience Australia Web Service, which offers datasets related to National Oil and Gas Infrastructure datasets [75]. The datasets include spatial positions of onshore gas and oil pipelines for the transmission of gas and oil all over the Australia mainland. The dataset also includes the position of gas and oil platforms across the territorial waters of Australia. The data from offshore gas platforms are compiled after gathering information from the Office of Transport Security (OTS). The raw data was uploaded to the ArcSDE environment which has geographical datasets of various types held in a relational Database system using FME (Feature Manipulation Engine). Geoscience Australia extracted the themed feature and then translated it into a new schema. The dataset was last revised on 1 January 2016. It has 372 nodes, including the receiving and connection points of gas pipelines featuring their diameter, shape length, owner, and construction year where we identified 212 edges within the gas transmission system.

## 5. Results and Discussion

This section discusses and compares the application of the previously mentioned complex network theories on the collected gas and electricity dataset of Australia. The feature analysis includes node degree distribution, fitting of the dataset, betweenness centrality, eigenvector centrality, and attributes such as the highest degree node, core, and connected components accordingly. For the model implementation, we used the Anaconda Jupyter programming platform and for plotting, we used the Python plotting package, matplotlib. In the following sections, we have highlighted important graphical representations and interpretations of the findings based on the supporting theories.

### 5.1. Topological Features

There are established approaches such as betweenness centrality or the Monte Carlo method that analyse system reliability by removing the components from the system to evaluate the consequences under different strategies of removal [28]. The concept is that the network can be resilient on holding certain components until the most important element or combination of some elements are removed leading to the whole network collapse [26]. In this report, the robustness to cascading failures will be analysed applying nodes and edges as removal elements using topological features and component characteristics.

Degree distribution: Figure 3 shows the degree distribution of the Australian gas and electricity networks, $N_k = \{n \in G: d(n) = k\}$.

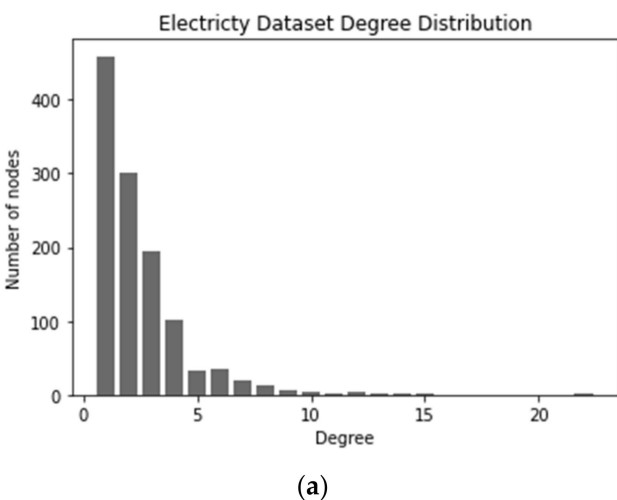

(a)

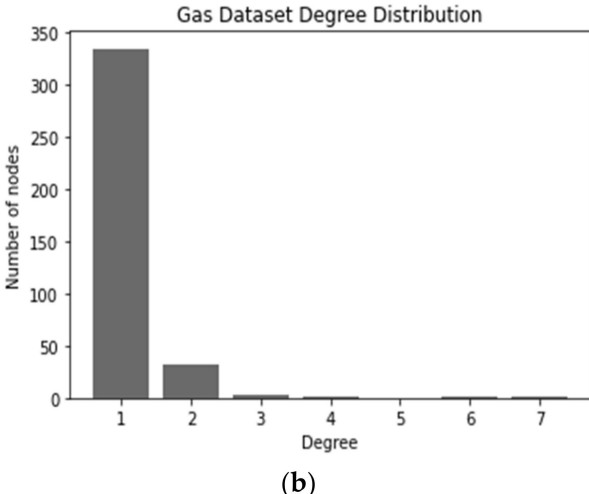

(b)

**Figure 3.** Node degree distribution of (**a**) electricity and (**b**) gas network datasets.

For more accurate judgement on the best predicting model, we have conducted curve fitting for each dataset against four models including ER, GE, WS, and BA. The results for the electricity and gas networks are illustrated in Figures 4 and 5, respectively. These figures clearly show that both the Barabasi-Albert and Erdos-Renyi models are the best to predict electricity networks, with the Barabasi-Albert model more suitable to represent the gas network. It can therefore be concluded that both gas and electricity networks can be considered scale-free.

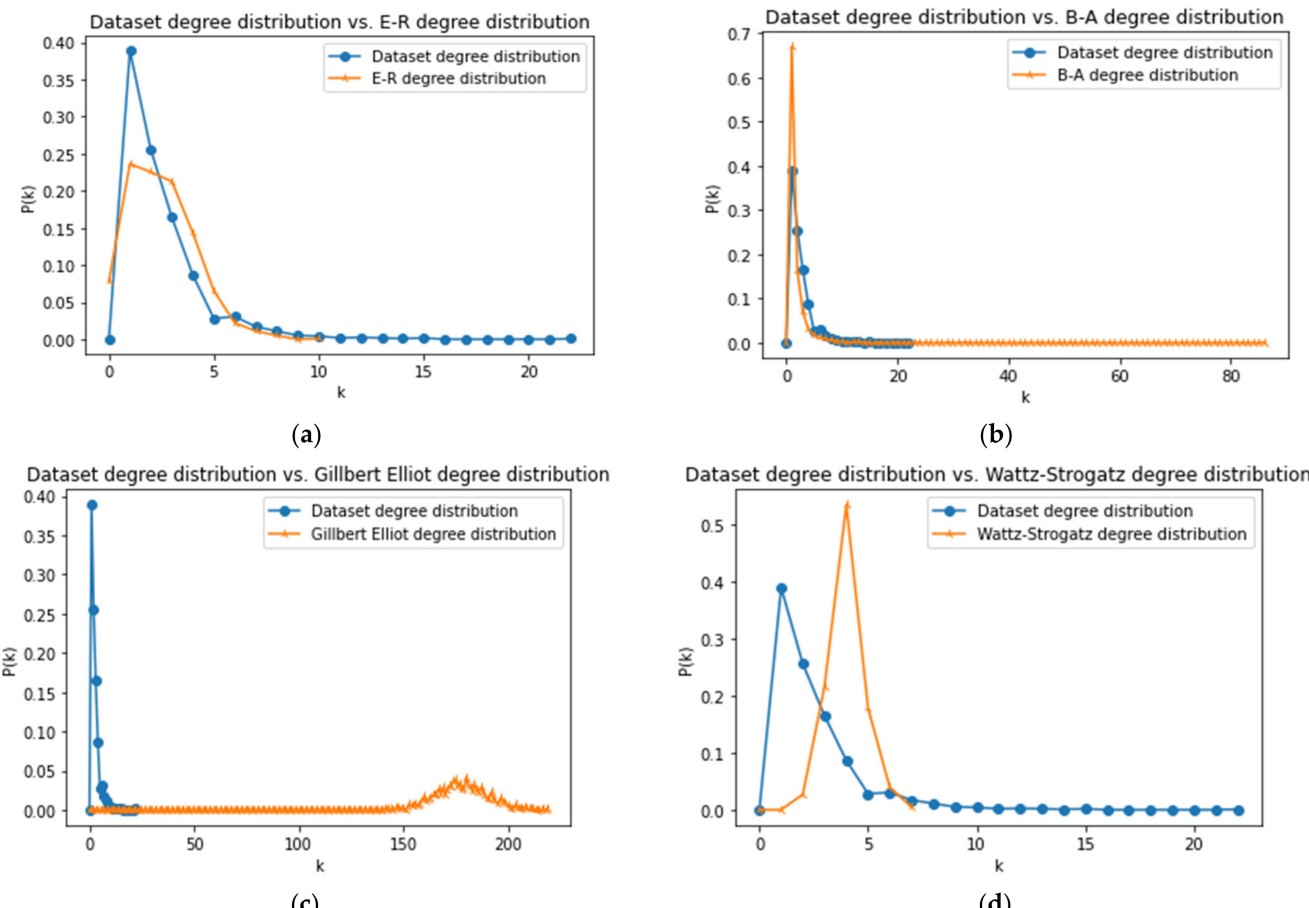

**Figure 4.** Fitness test of electricity network degree distribution with four network models: (**a**) ER, (**b**) BA, (**c**) GE, and (**d**) WS.

Centrality measures: We have studied two centrality measures including betweenness and eigenvectors. Betweenness centrality is the fraction of the shortest paths between any pair of nodes $s$ and $t$ that passes through a node $v$, given by $g(v) = \sum_{s \neq v \neq t} \sigma_{st}(v)/\sigma_{st}$, where $\sigma_{st}(v)$ is the total number of paths from any node $s$ to any node $t$ through $v$, $\sigma_{st}$ is the total number of paths from any node $s$ to any node $t$, and $g(v)$ = betweenness centrality value of the node $v$. The betweenness centrality of gas and electricity networks are illustrated in Figure 6. For the electricity network, the highest betweenness centrality value is 0.427, with 1117 of the total 1180 nodes (i.e., 94.7%) having values less than 0.042. This is also an implication of a scale-free network with a tail (hubs). A similar trend is observed for the gas network, with 364 of the total 372 nodes (i.e., 97.8%) having betweenness centrality values less than 0.0005. The other 8 nodes have centrality values in the range of 0.0005–0.005.

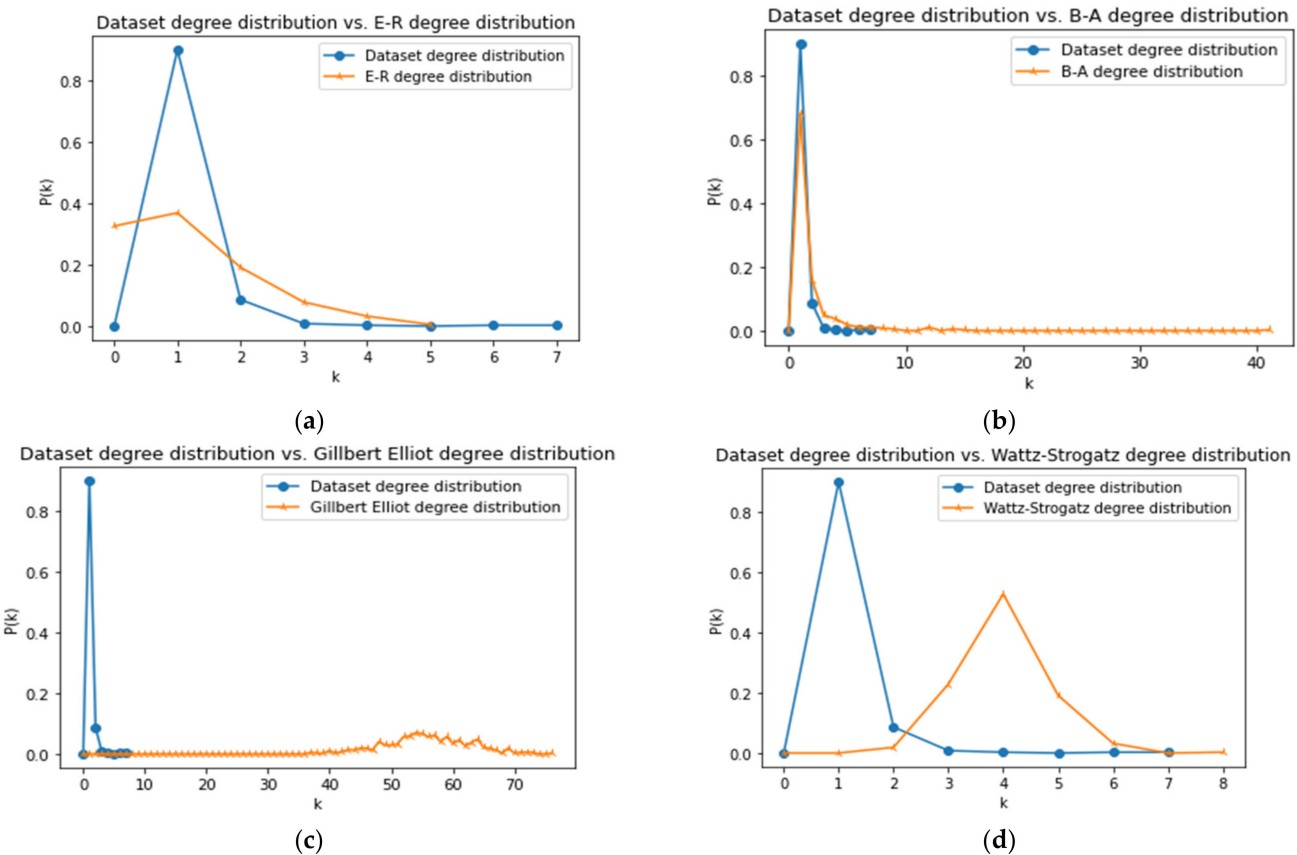

**Figure 5.** Fitness test of gas network degree distribution with four network models: (**a**) ER, (**b**) BA, (**c**) GE, and (**d**) WS.

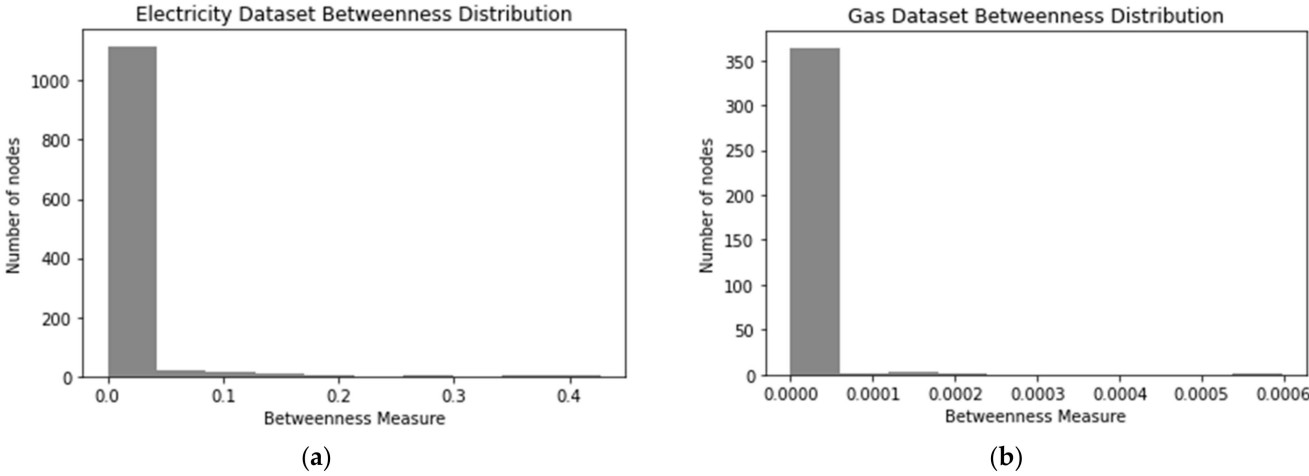

**Figure 6.** Betweenness centrality distribution experiment on Australian (**a**) electricity and (**b**) gas network datasets.

Eigenvector centrality provides a more in-depth view of the centrality measure where the value assigned to a node is based on its influence on the whole network. Initially, all nodes are considered to have the same level of influence on the network but with the progress in computing, the nodes with higher degree values have a leveraging influence. The computation iterates continually by assigning scores to all nodes until the final set of scores for each node is stabilised.

The results of the eigenvector centrality of gas and electricity networks are illustrated in Figure 7. For the electricity network, 1147 of the total 1180 nodes (i.e., 97.2%) have eigenvector centrality values less than 0.056. Another 32 nodes have values in the range of 0.056–0.227, with only 1 node having a value between 0.512 and 0.569. For the gas network, 362 of the total 372 nodes (i.e., 97.3%) have eigenvalues less than 0.068 and 9 nodes are in the 0.068–0.549 range. The value of the node with the highest eigenvalue is 0.686.

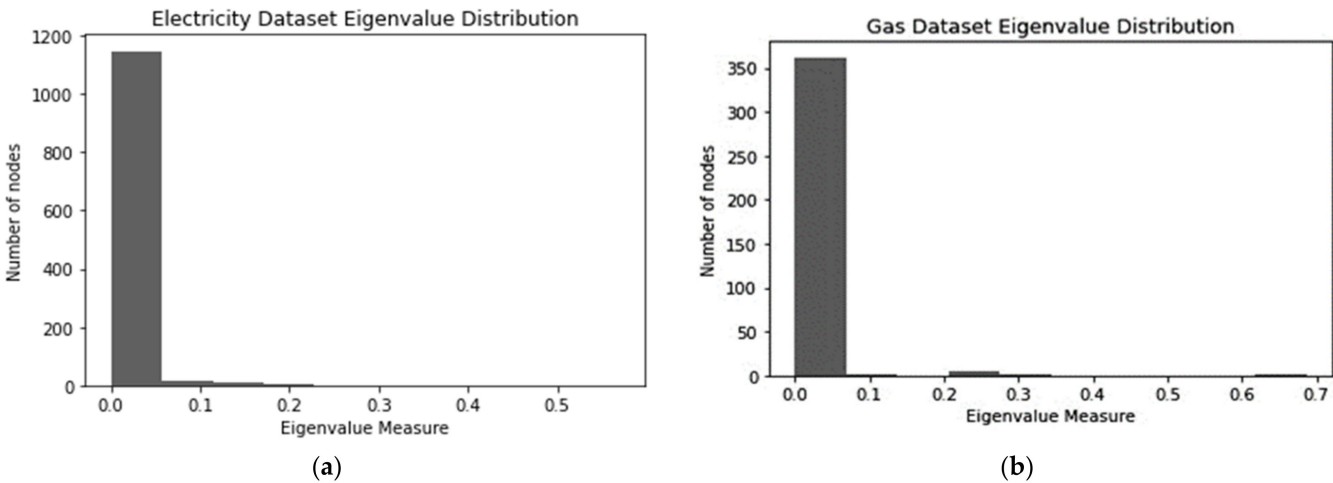

(**a**)                                                                        (**b**)

**Figure 7.** Eigenvector centrality distribution experiment on Australian (**a**) electricity and (**b**) gas network datasets.

The mean value of eigenvalue measures for the electricity network is 0.0070095714, with the median being $5.44169 \times 10^{-6}$. For the gas network, the mean value of eigenvalue measures is 0.0076266264, with a median (eigenvalue measure) of $1.0135194 \times 10^{-23}$.

Implications of centrality results: The results summary for both electricity and gas networks is provided in Table 1. The centrality analysis shows that the highest degree node is node 293, which is a substation at NSW with a degree of 22. This substation also has the highest betweenness centrality which reaffirms its hub feature. For the gas dataset, the highest degree node is located in Victoria, with a degree of 7 within a major demand centre.

**Table 1.** Summary of highest nodes, core, and number of connected components within two networks.

|  | **Electricity Network** | **Gas Network** |
| --- | --- | --- |
| Highest degree node | Node 293 | Node "Ddg" |
| Degree of highest degree node | 22 | 7 |
| Number of nodes in core (% of total nodes) | 1092 nodes (92.54%) | 11 nodes (2.95%) |
| Number of connected components | 42 | 160 |

Core measures: The core of a network refers to the set of nodes in the largest connected component of the network. The largest connected component of the electricity network has a count of 1092 nodes (92.54% of the total 1180 nodes) which means that all these 1092 nodes are connected (Figure 8a). A deeper analysis using the Breadth-First Search Algorithm (BFS) estimated that there were 42 connected components in the network. In the case of the gas dataset, the core of the connected components has relatively fewer nodes compared to the electricity dataset. The largest connected component of the gas dataset had a count of just 11 nodes (2.95% of the total 372 nodes), but the number of connected components in the network was estimated to be 160 (Figure 8b). Thereby, we can understand that there are more independently connected components in the gas network compared to the electricity network.

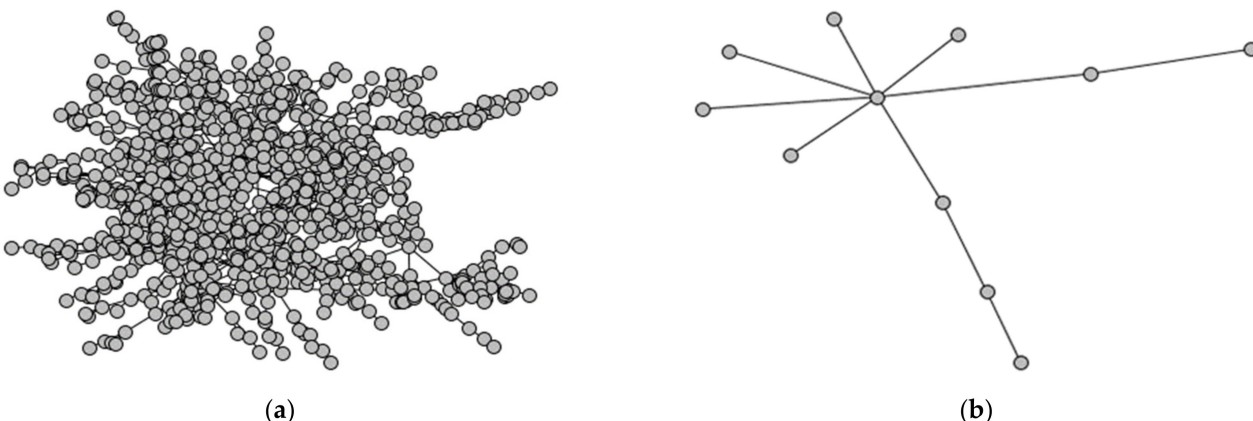

(**a**)          (**b**)

**Figure 8.** The largest connected component of (**a**) electricity and (**b**) gas networks.

*5.2. Resilience Analysis against Fault or Failure*

In the previous section, we studied various network topology measures such as degree distribution and centrality. Here, the aim is to study the impact of fault or attack following any of those discussed measures. We achieve this by conducting node and edge removal experiments on the Australian electricity and gas dataset.

Random failure versus targeted attack: The robustness analysis was performed to understand the vulnerability of the electricity network and to estimate which kind of attack would make the network more prone to collapse. The removal of nodes was performed by inducing four kinds of attacks: (1) random failure, (2) degree-based attack, (3) betweenness-based attack, and (4) eigenvector-based attack, which are represented in Figure 9 (for electricity network) and Figure 10 (for gas network). For the electricity network, Figure 10 shows that the curve representing degree-based attack drops quicker than the one representing random failure, indicating that the network is prone to collapse quicker from a degree-based attack than a random failure. The robustness examined by the factor of the proportion of nodes in the core indicates that the network offers more resistance to random-based failure of nodes than intentional or targeted attacks. Furthermore, when comparing betweenness and eigenvalue centrality attacks, it can be observed that the network topples relatively quicker in a betweenness centrality attack than an eigenvalue centrality attack. This can be explained by the fact that the betweenness centrality attack is based on a targeted single node while the eigenvalue centrality attack focuses on a cluster of nodes which is relatively more difficult to break. A further comparison of single-node targeted attacks with both degree-based and betweenness centrality approaches reveals that the proportion of nodes in the core during the degree-based attack is less compared to that of the betweenness centrality-based attack. The proportion of nodes in the core is still approximately 0.3 after the removal of the highest betweenness centrality node but the proportion of nodes drops to zero after removing another node. Thus, the electricity network is least resilient if subjected to degree-based attacks.

The robustness analysis involving the same attacks as that of the electricity network was performed on the gas network. As shown in Figure 10, the robustness examined by the factor of a proportion of nodes in the core indicates that the network offers more resistance to randomly based failure of nodes than intentional or targeted attacks. In terms of betweenness and eigenvalue centrality attacks, it can be observed that the network topples relatively quicker in a betweenness centrality attack. This is for the same reason as explained in the electricity network, where the betweenness centrality attack is based on a single node while the eigenvalue centrality attack is based on high scores computed for a cluster of nodes.

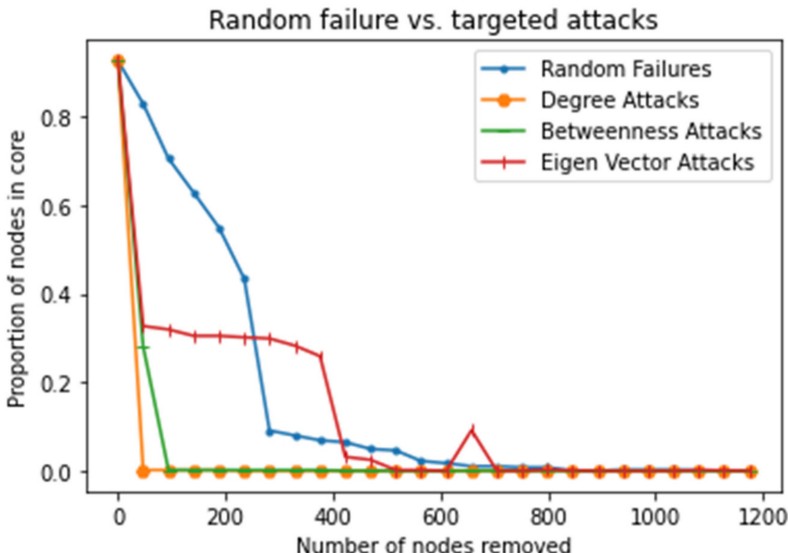

**Figure 9.** The resilience of the Australian electricity network to random failure or targeted attacks on the nodes (generators or substations) following degree-based, betweenness-based, and eigenvector-based strategies.

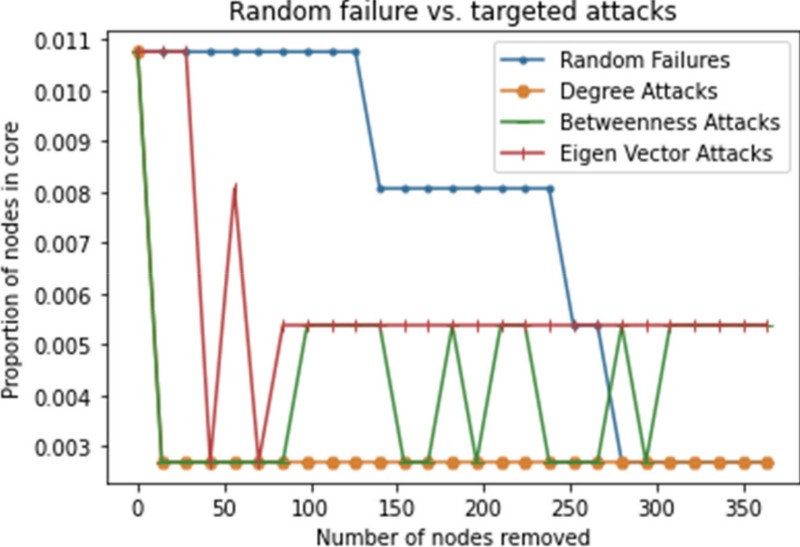

**Figure 10.** The resilience of the Australian gas network to random failure or targeted attacks on the nodes (generators or substations) following degree-based, betweenness-based, and eigenvector-based strategies.

However, on further comparing the single node-based attacks, it can be observed that the proportion of nodes in the core falls to zero and never increases during a degree-based attack compared to random failures and betweenness and eigenvalue centrality attacks. Compared to the electricity network, the gas network is more resistant to degree-based attack as it requires the removal of at least two of the highest degree nodes to collapse. Thus, the gas network is also least resilient if subjected to degree-based attacks.

Comparing the analysis on the electricity (Figure 9) and gas (Figure 10) networks, one interesting observation in the gas network is that the proportion of nodes tends to increase in both betweenness and eigenvalue centrality attacks. This is because the number of nodes constituted in the core (largest connected component) is far less compared to the core of the electricity dataset, plus the gas network has a larger number of individual connected components than the electricity network. Since the core has only 11 nodes, once the attack removes all the nodes from the core, then the second-largest component becomes the core

which has a certain proportion of nodes existing. This kind of observation is not found in the electricity network as the core itself had most of the nodes (1092) constituted.

Node and edge removal: Every network is comprised of nodes and edges. Threats could occur to a network in any form and on any part of the network, including node or edge. In particular, critical infrastructures can be subject to threats on transmission lines. The criteria of random edge-based removal and node-based removal were taken into consideration to analyse the robustness of the network.

The comparison was made to estimate whether the network is more resilient to node-based removal or edge-based removal. The analysis was made with respect to what kind of effect it is going to have with the proportion of edges or nodes in the core by randomly removing them. Based on the analysis, it was understood that the electricity network had more resistance to edge-based removal than node-based removal (Figure 11).

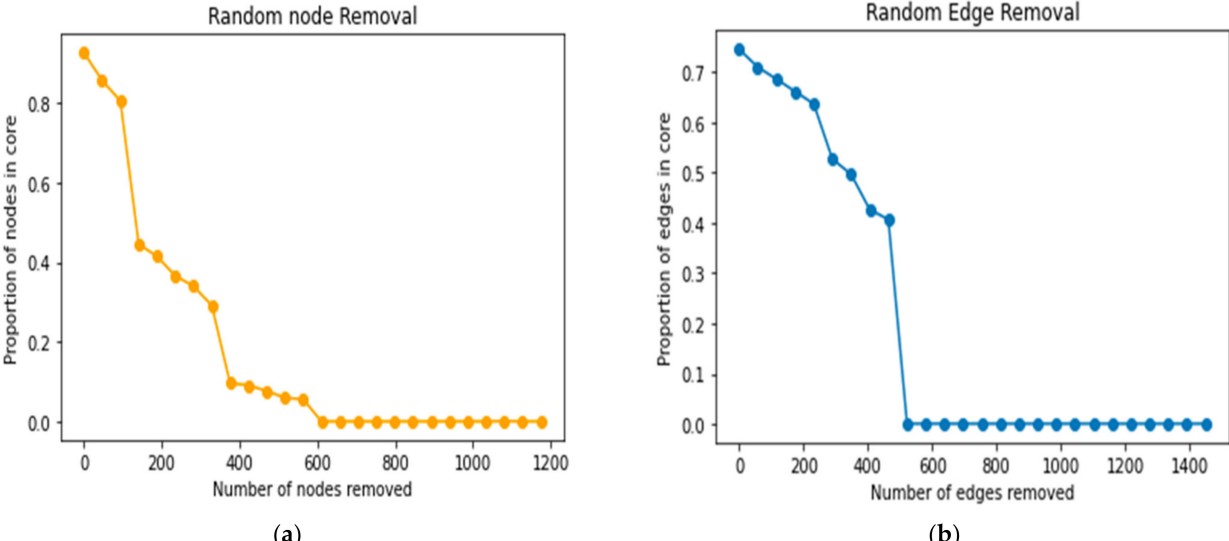

**Figure 11.** (**a**) Node and (**b**) edge removal experiment on an Australian electricity network dataset.

A significant number of edges need to be removed from the core for the network to collapse (blue line) but the removal of just a few nodes can disrupt the whole network (orange line). During random node removal, as shown in Figure 11a, the network collapses after the removal of 600 nodes. During random edge removal, as shown in Figure 11b, almost 500 edges have to be removed for the network to collapse, which is equivalent to approximately 1000 nodes (one edge comprises two nodes). Thus, this visualisation helps in illustrating that the network is more influenced by node-based removal than edge-based removal. Random attacks in removing edges are thereby less effective compared to random attacks in removing nodes.

A similar phenomenon was observed while analysing the gas dataset, where a significant number of edges has to be removed for the network to completely collapse but the removal of few nodes can make the network collapse at a faster rate (refer to Figure 12).

This indicates that a node-based attack also influences the gas network more than edge-based removal, because removing a node leads to removing the edges connected to that node as well. Thus, the robustness of the gas network is also influenced comparatively by random node-based removal than the random edge-based removal.

Weight-based edge removal versus unweighted-based edge removal: Earlier analysis included only the topological attributes without including the physical attributes of the network graph. The physical attributes include measures such as the cable capacity, the frequency of transmission, or the physical length of the cable. In the case of the electricity dataset, the parameter of the capacity of transmission, "voltage", was considered, while for the gas dataset, the physical length of the cable was considered.

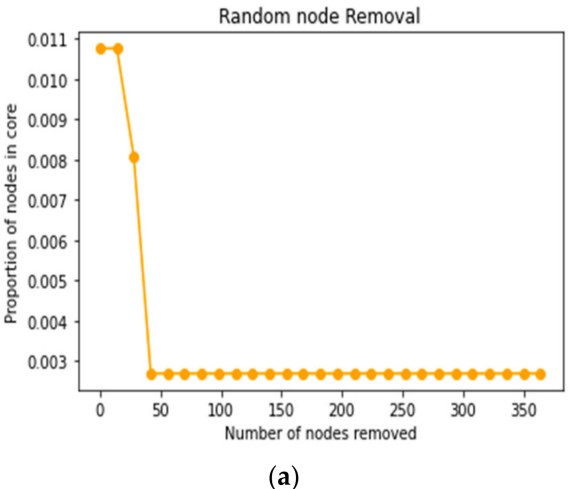
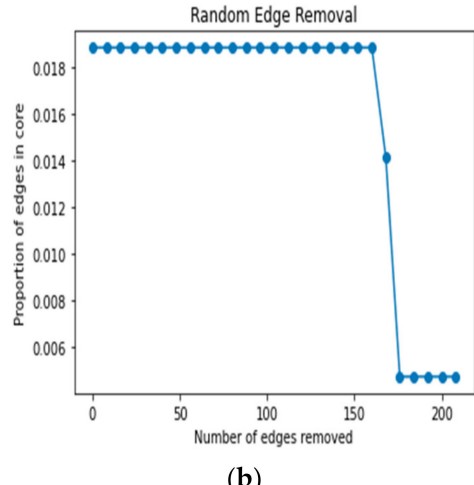

(**a**)  (**b**)

**Figure 12.** (**a**) Node and (**b**) edge removal experiment on an Australian gas network dataset.

Figure 13 shows the analysis results of the weight-based edge removal experiment for the electricity network, using the voltage between two substations as the weight. The edges were sorted by weight in descending order, with the removal of edges completed considering the weight parameter which, in this case, was voltage. The removal of nodes without considering weight makes the network collapse quickly when the critical node is removed.

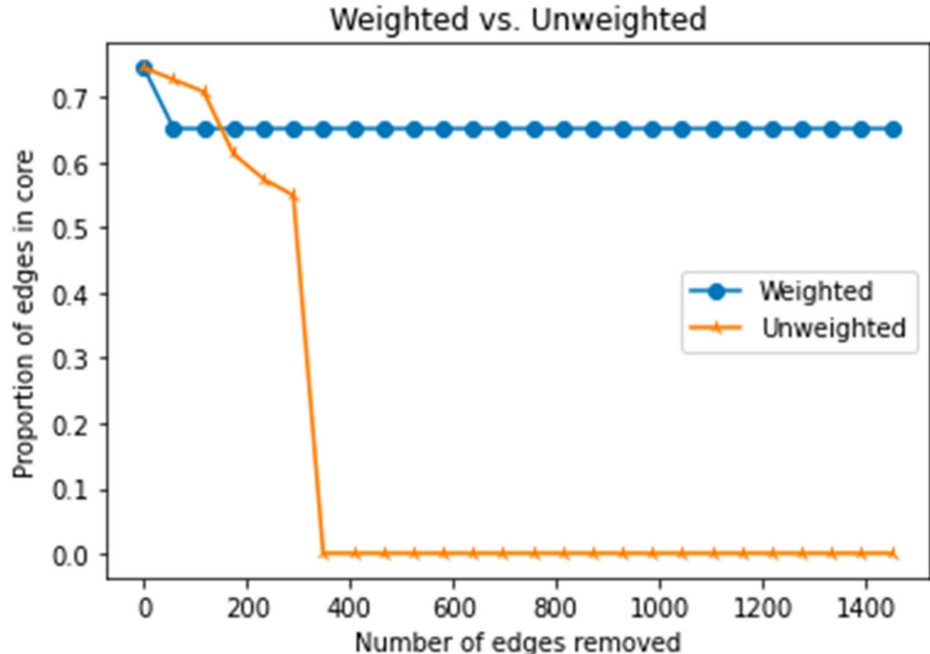

**Figure 13.** Weight-based edge removal experiment on an Australian electricity transmission network dataset.

Similarly, Figure 14 shows the analysis results of the weight-based edge removal experiment for the gas network, using the length between two transmitting pipelines as the weight. A similar kind of phenomenon was observed in the analysis of the gas network, where a higher number of significant nodes have to be removed from the core during weight-based removal for the network to collapse.

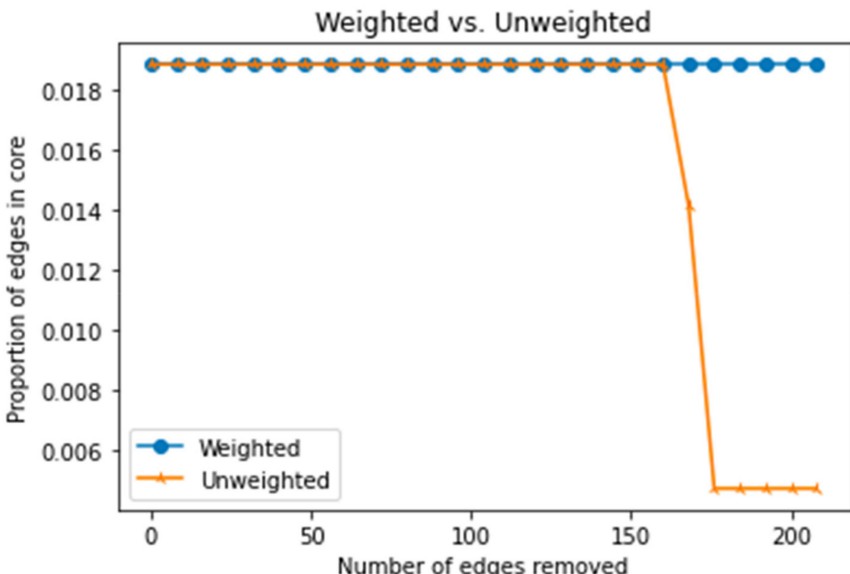

**Figure 14.** Weight-based edge removal experiment on Australian gas transmission network.

## 6. Conclusions

This paper focused on Australia's critical infrastructure energy networks, including electricity and gas. Any disturbance to these networks can cause multidimensional catastrophe. Understanding the characteristics and behaviour of these networks helps the relevant stakeholders to assess network resilience under various scenarios. This makes them proactive about certain situations, delivering a resolution to failures, creating mechanisms to balance supply with demand following the continuously growing population, and enhancing quality of life with innovative technology.

In this report, we utilised theories of network science to study the characteristics of the Australian gas and electricity networks and further explore their features to examine the robustness of these networks. We tested the two networks against four models: Random Graph model, which includes the (I) Erdős-Rényi model and the (II) Gilbert-Elliott model; (III) Scale-Free Network model, which includes the Barabási-Albert (BA) model; and (IV) Watts-Strogatz model, which includes the Small World Network model.

In summary, this research modelling and its outcome started by collecting datasets from Geoscience Australia Web Service, cleaning them for better clarity, and analysing them using the Python NetworkX package. The analysis revealed both the Australian gas and electricity networks as scale-free networks. Then, different models built to examine the network robustness demonstrated that both networks are more robust to random errors and sensitive to intentional attacks. Additionally, the networks exhibit the least resistance to degree-based attacks, followed by betweenness centrality-based attacks, eigenvalue centrality-based attacks, and the most resistance to random failures.

This illustration is significant in understanding how robust the Australian electricity and gas network is and the various insights in preventing these networks' catastrophic failures. However, certain assumptions were made in terms of analysis and which on incorporating, could have brought greater finetuning of the analysis.

Future works: Our present analysis was based on the overall Australian gas and electricity dataset, but it would be interesting to undertake an in-depth study on highly clustered network zones using related voltage or gas capacity data. This would help focus on the most critical zones to identify the vital nature of the cluster and how vulnerable it could be to cascading failure. For the gas network, we only used the receiving and delivering points of the pipelines as edges or nodes, with the robustness of weighted and unweighted networks achieved based on limited parameters. However, further work is needed to assess whether any correlation of the critical nodes exists with parameters such as pipeline diameter or age.

**Author Contributions:** Conceptualization, S.A.K., M.T., Z.S.B., F.K. and K.K.; methodology, S.A.K., M.T., Z.S.B. and K.K.; software, S.A.K.; validation, S.A.K.; formal analysis, S.A.K., M.T., Z.S.B. and K.K.; investigation, S.A.K., M.T., Z.S.B., F.K. and K.K.; data curation, S.A.K., M.T., Z.S.B. and K.K.; writing—original draft preparation, S.A.K., M.T. and Z.S.B.; writing—review and editing, S.A.K., M.T., Z.S.B., F.K. and K.K.; visualization, S.A.K. and Z.S.B.; supervision, K.K.; project administration, K.K. All authors have read and agreed to the published version of the manuscript.

**Funding:** This research received no external funding.

**Institutional Review Board Statement:** This study used secondary data and ethical approval was not required.

**Informed Consent Statement:** Not applicable.

**Data Availability Statement:** Open source data was used in this study with the reference cited.

**Conflicts of Interest:** The authors declare no conflict of interest.

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
