# Peer review of "Resilience Analysis of Australian Electricity and Gas Transmission Networks"

_sustainability, doi:10.3390/su14063273_

Round 1

Reviewer 1 Report

This manuscript discusses an important sustainability subject related to the resilience of the electricity and gas distribution networks in the country of Australia. The approach used for analyzing the reliability in such networks is traditional with no apparent novelty. However, the findings and conclusions based on the collected data are important. There are some concerns that need to be addressed as provided in the following points.

  • In the abstract, it seems awkward to talk about “intentional attacks” in a scientific research paper that is not concerned with international relations and/or conflicts. The reviewer suggests focusing on natural disasters and internal socio-political problems that might result in the stated attacks in the manuscript.
  • The manuscript covers sufficient background related to the resilience analysis in complex distribution networks. However, it is not clearly stated which theory or method is adopted in the conducted analysis of the case study.
  • In line 366, “ArcSDE” is not defined. Please provide appropriate reference.
  • The date of the last revision of the data is stated to be January 1, 2016, which is now more than 6 years old. A reader might wonder how the collected data reflects the current situation and whether the conducted research is relevant as of today or not. Please explain.
  • In lines 454-456, it is written “A deeper analysis using Breadth-First Search Algorithm (BFS) led to the estimation of the number of connected components in the network which was found to be 42 in number.” It is not clear how the application of the BFS algorithm is conducted to lead to 42 connected nodes as opposed to the 1092 nodes illustrated in Figure 8a. Please explain.
  • The Authors contributions section starting in line 623 is not related to the submitted manuscript. Please rewrite.

Author Response

We sincerely appreciate the very constructive comments. We have attempted to address them to the maximum extent possible.

Reviewer #1: This manuscript discusses an important sustainability subject related to the resilience of the electricity and gas distribution networks in the country of Australia. The approach used for analyzing the reliability in such networks is traditional with no apparent novelty. However, the findings and conclusions based on the collected data are important. There are some concerns that need to be addressed as provided in the following points.

  • In the abstract, it seems awkward to talk about “intentional attacks” in a scientific research paper that is not concerned with international relations and/or conflicts. The reviewer suggests focusing on natural disasters and internal socio-political problems that might result in the stated attacks in the manuscript.

Response: Thank you. Intentional attack (or sabotage) is a common term in the network resilience study. However, to address this comment, we have changed the “intentional attack” to “sabotage”. The natural disaster is a valid term that has come soon after this sentence. The abstract now reads:

“As two critical infrastructures, the security of electricity and gas networks is highly important due to potential multifaceted social and economic impacts. Unexpected errors or intentional attackssabotage can lead to blackouts causing a significant loss for the public, businesses and governments. Climate change and the increasing bushfires and floodsnatural disaters, e.g., booshfires and floods, are other emerging network resilience challenges.”   

  • The manuscript covers sufficient background related to the resilience analysis in complex distribution networks. However, it is not clearly stated which theory or method is adopted in the conducted analysis of the case study.

Response: Theories related to Graph theory models were adopted to analyse the case study under Section 5.1. Figure 4 and figure 5 represents how different graph theory models fit to the dataset illustrated in the case study. On the basis of our observations, we figured out that the optimal graph theory model which closely aligns with the dataset is the Barabasi Albert Model.

  • In line 366, “ArcSDE” is not defined.

Response: Explanation is provided in the text. 

  • The date of the last revision of the data is stated to be January 1, 2016, which is now more than 6 years old. A reader might wonder how the collected data reflects the current situation and whether the conducted research is relevant as of today or not. Please explain.

Response: These kind of data are not provided frequently. We have used the latest available data. During our study we contacted some national organisations for updated data and noticed they use also the same 2016 data in their analyses.

  • In lines 454-456, it is written “A deeper analysis using Breadth-First Search Algorithm (BFS) led to the estimation of the number of connected components in the network which was found to be 42 in number.” It is not clear how the application of the BFS algorithm is conducted to lead to 42 connected nodes as opposed to the 1092 nodes illustrated in Figure 8a. Please explain.

Response: The connected components were generated using a python pre-defined function called ‘connected_components (G)’ where G is the undirected graph. This pre-defined function connected_components (G) will generate the list of connected components and the backend algorithm involved for generating this connected component is BFS. Therefore, when the pre-defined function is executed, then it outputs the list of connected components, and the entire length of the list was estimated to be 42.

No of nodes in largest connected component = 1092

No of connected components = 42

  • The Authors contributions section starting in line 623 is not related to the submitted manuscript. Please rewrite.

Response: Thank you for noticing this. The first two lines were from the word template. But, the remainder were about this paper. We have removed the redundant lines.

Reviewer 2 Report

Summary

In this paper, the authors study the vulnerability of electricity and gas supply networks to outages and problems due to the failure of network nodes or arcs. The authors carry out an analysis of the network topology and determine the distribution that best adapts to the different networks analysed. The article analyses the network study of two networks: an electricity network and a gas network. Both networks are on the east coast of Australia. The main conclusions are that both networks are scale-free networks and are more robust to the occurrence of random errors against intentional attacks. 

Comments

Overall, the paper is correct and well-written. Conclusions sounds according the analyses and previous results. Authors have included a review of the previous literature and a brief reference to some real examples of failures in electricity and gas networks around the World.

The first part of the work includes a study of the resilience of power networks (electricity, gas, ...) to the occurrence of faults or power cuts. These problems can have a random or intentional origin. Several real examples of these situations are discussed. On the other hand, they focus their work on Australia considering gas and electricity networks, including several references to previous works on the same subject. They then provide a good review of the literature on this topic, focusing on network analysis models, random graph models, free-scale network models and small-world network models. Finally, in this first part, they consider the study of network robustness as a function of the micro and macro characteristics of the network and the percolation theory that analyses how the elements of a network relate to each other.

The second part of their work considers the case study of the electricity and gas network of the south-east coast of Australia. Based on the available data, the authors carry out (section 5) a topological analysis of the network and determine the different measures considered in the analysis: degree distribution, centrality measures (betweenness centrality and eigenvector centrality), number of nodes in core and number of connected components. And finally, they analyse the resilience of the two networks (electricity and gas) to random failures and intentional attacks, considering three types of attacks: degree attacks, betweenness attacks and eigenvectors attacks.

All the work is correct and the results are consistent with the theory and analysis considered. The work is simple and correct although it does not go into depth on several aspects such as:

- Considering the network as a directed graph, and not as is done in the paper as an undirected graph, which allows arcs to be used in any direction.

- To study a weighting for the network. In such a way that the fall of a node is not only important for the number of arcs but also for its usefulness in the network (primary supply, secondary supply, etc.)

Although some of these ideas are suggested as possible lines of research in future work.

It is a good paper, where an interesting application of the study of networks and graphs to a real situation is presented. And from the theoretical study of the network, several conclusions are obtained about the resilience and robustness of these networks in the face of network outage problems.

Specific Comments

Some specifics questions to consider:

  1. Throughout the work the authors use two systems for referencing the bibliography in the text. In order to improve the readability of the work, it would be advisable to unify the citation system used for bibliographical references.
  2. Page 4, line 133. There is an extra parenthesis.
  3. Page 6, line 231. You must use subscripts, where it required.
  4. Page 6, line 235. Revise the las expression of this line. You must define all the variables. And you might use the symbol >> (much greater than) when it required.

Author Response

We sincerely appreciate the very constructive comments. We have attempted to address them to the maximum extent possible.

Comments

Overall, the paper is correct and well-written. Conclusions sounds according the analyses and previous results. Authors have included a review of the previous literature and a brief reference to some real examples of failures in electricity and gas networks around the World.

The first part of the work includes a study of the resilience of power networks (electricity, gas, ...) to the occurrence of faults or power cuts. These problems can have a random or intentional origin. Several real examples of these situations are discussed. On the other hand, they focus their work on Australia considering gas and electricity networks, including several references to previous works on the same subject. They then provide a good review of the literature on this topic, focusing on network analysis models, random graph models, free-scale network models and small-world network models. Finally, in this first part, they consider the study of network robustness as a function of the micro and macro characteristics of the network and the percolation theory that analyses how the elements of a network relate to each other.

The second part of their work considers the case study of the electricity and gas network of the south-east coast of Australia. Based on the available data, the authors carry out (section 5) a topological analysis of the network and determine the different measures considered in the analysis: degree distribution, centrality measures (betweenness centrality and eigenvector centrality), number of nodes in core and number of connected components. And finally, they analyse the resilience of the two networks (electricity and gas) to random failures and intentional attacks, considering three types of attacks: degree attacks, betweenness attacks and eigenvectors attacks.

All the work is correct and the results are consistent with the theory and analysis considered. The work is simple and correct although it does not go into depth on several aspects such as:

- Considering the network as a directed graph, and not as is done in the paper as an undirected graph, which allows arcs to be used in any direction.

Response: For the current study, we have done the analysis considering the network to be an undirected but weighted graph. The direction issue in a large-scale electricity network is not straightforward. There are some transmission lines which transfers power in one direction at one point of time and transfers reversely at another point of time.   As such, very often national level resilience analyses are carried out on the basis of undirected network. For instance, the well-recognised review paper titled “The Power Grid as a complex network: A survey” by Paganini and Aiello (2013) states that “Another main commonality is to treat the Grid as an undirected graph where each substation or transformer represents a node and each line transporting electricity is an edge.”

Pagani, G. A., & Aiello, M. (2013). The Power Grid as a complex network: A survey. Physica A: Statistical Mechanics and Its Applications, 392(11), 2688–2700. https://doi.org/https://doi.org/10.1016/j.physa.2013.01.023

It is a good paper, where an interesting application of the study of networks and graphs to a real situation is presented. And from the theoretical study of the network, several conclusions are obtained about the resilience and robustness of these networks in the face of network outage problems.

Response: Thank you.

Specific Comments

Some specifics questions to consider:

  1. Throughout the work the authors use two systems for referencing the bibliography in the text. In order to improve the readability of the work, it would be advisable to unify the citation system used for bibliographical references.

Response: So far we have used Mendeley for deriving referencing.

  1. Page 4, line 133. There is an extra parenthesis.

Response: Extra parenthesis removed.

  1. Page 6, line 231. You must use subscripts, where it required.

Response: Subscripts have been fixed.

  1. Page 6, line 235. Revise the las expression of this line. You must define all the variables. And you might use the symbol >> (much greater than) when it required.

Response: somehow the equation got changed while editing. Fixed as commented.

Reviewer 3 Report

The paper analyses the resilience of Australian electricity and gas transmission networks. It is written in a very understandable language in a clear and distinct way, which I highly appreciate. The aim of this work, background and used methods, as well as results are described in a high quality. From my perspective the paper is fine, I have just a few comments:

  • Some small typing errors (lines 205, 355, 573)
  • Figure 7 is the same as figure 6.
  • Paragraph line 570-575: I am not sure, if I understood this paragraph right. The voltage drop is meant by the “voltage between two substations”, right? Do I understand the text correct, that the “descending order” mean, rather lines with higher voltage drops are removed?
  • At the end of chapter 5 I would be really curious to read an assessment of the results.

Author Response

We sincerely appreciate the very constructive comments. We have attempted to address them to the maximum extent possible.

The response file is enclosed.

Reviewer 4 Report

My deep conviction that the current level of development of science makes it very topical to consider models and results obtained for biological, social sciences in engineering problems and vice versa. The key to obtaining new solutions is in a comprehensive study of the problem with the involvement of a mathematical apparatus that have had a successful application experience in a number of areas. The very idea of examining models that have shown their effectiveness in a certain area to a new subject area deserves attention. Even if the final result does not turn out to be better than the previous ones, the research methodology provides new ideas for developing models. Of course, the correctness of transferring models to a new subject area should be justified. The authors demonstrate the application of models developed for the World Wide Web, social networks, citation network, etc. to the Australian gas and electricity networks. It is worth noting that such networks are very specific to each country, so it would be interesting to see and compare the results of a similar study for several other countries.

The biggest questions when reading the article arose about the validity of the application of the methods, given that the electric and gas networks are represented by directed graphs with a very small number of sources and a large number of consumers. Further development of electricity and gas supply systems will lead to their sharp difference, since when using alternative energy sources in electricity supply, the nature of the grouping of connections in the graph will change, while the existing gas networks will continue to be used, retaining their current topology. 

Below I formulate 2 questions that may apply not only to the indicated lines, but also to some paragraphs where the corresponding models are mentioned:

1. Lines 237-238 The probability distribution of the three network models: For example, it is stated that “The degree distribution of small-world networks (Wattz Strogatz Model) follows the exponential function”

If we consider a directed graph, would this be true?

2.  Lines 325-326 What features will this approach have when working with directed graphs, taking into account the limited ability of sources?

As a conclusion: the presented article is original, focused on solving important problems, corresponds to the aims of Sustainability, I recommend to accept it after minor revision

Author Response

We sincerely appreciate the very constructive comments. We have attempted to address them to the maximum extent possible.

Reviewer #4:

My deep conviction that the current level of development of science makes it very topical to consider models and results obtained for biological, social sciences in engineering problems and vice versa. The key to obtaining new solutions is in a comprehensive study of the problem with the involvement of a mathematical apparatus that have had a successful application experience in a number of areas. The very idea of examining models that have shown their effectiveness in a certain area to a new subject area deserves attention. Even if the final result does not turn out to be better than the previous ones, the research methodology provides new ideas for developing models. Of course, the correctness of transferring models to a new subject area should be justified. The authors demonstrate the application of models developed for the World Wide Web, social networks, citation network, etc. to the Australian gas and electricity networks. It is worth noting that such networks are very specific to each country, so it would be interesting to see and compare the results of a similar study for several other countries.

The biggest questions when reading the article arose about the validity of the application of the methods, given that the electric and gas networks are represented by directed graphs with a very small number of sources and a large number of consumers. Further development of electricity and gas supply systems will lead to their sharp difference, since when using alternative energy sources in electricity supply, the nature of the grouping of connections in the graph will change, while the existing gas networks will continue to be used, retaining their current topology.

Below I formulate 2 questions that may apply not only to the indicated lines, but also to some paragraphs where the corresponding models are mentioned:

  1. Lines 237-238 The probability distribution of the three network models: For example, it is stated that “The degree distribution of small-world networks (Wattz Strogatz Model) follows the exponential function”

If we consider a directed graph, would this be true?

Response: The electricity networks cannot be completely interpreted as a direct graph network, as at one point of time if the power may flow in one direction, then at some other point of time it might be flowing in the reverse direction.

The Wattz Strogatz Model depends on two parameters namely the average path length and clustering coefficient. In the case of directed graphs, the degree distribution would vary as the degree of a node in a directed graph would be the number of edges incident on the specified node whereas in an undirected graph the degree of a node would be the number of inter-connections it has. Therefore, there would be a change in the degree distribution which in turn affects the average path length and clustering coefficient and would not follow an exponential function.

  1. Lines 325-326 What features will this approach have when working with directed graphs, taking into account the limited ability of sources?

Response: For the current study, we have done the analysis considering the network to be an undirected but weighted graph. The direction issue in a large-scale electricity network is not straightforward. There are some transmission lines which transfers power in one direction at one point of time and transfers reversely at another point of time.   As such, very often national level resilience analyses are carried out on the basis of undirected network. For instance, the well-recognised review paper titled “The Power Grid as a complex network: A survey” by Paganini and Aiello (2013) states that “Another main commonality is to treat the Grid as an undirected graph where each substation or transformer represents a node and each line transporting electricity is an edge.”

Pagani, G. A., & Aiello, M. (2013). The Power Grid as a complex network: A survey. Physica A: Statistical Mechanics and Its Applications, 392(11), 2688–2700. https://doi.org/https://doi.org/10.1016/j.physa.2013.01.023

As a conclusion: the presented article is original, focused on solving important problems, corresponds to the aims of Sustainability, I recommend to accept it after minor revision

Response: Thank you.

Round 2

Reviewer 1 Report

The authors' responses and modifications are satisfactory.